# Nonstochastic Multiarmed Bandits
# with Unrestricted Delays

**Tobias Sommer Thune**[*]
University of Copenhagen
Copenhagen, Denmark
tobias.thune@di.ku.dk

**Nicolò Cesa-Bianchi**
DSRC & Univ. degli Studi di Milano
Milan, Italy
nicolo.cesa-bianchi@unimi.it

**Yevgeny Seldin**
University of Copenhagen
Copenhagen, Denmark
seldin@di.ku.dk

## Abstract

We investigate multiarmed bandits with delayed feedback, where the delays need neither be identical nor bounded. We first prove that "delayed" Exp3 achieves the $\mathcal{O}\big(\sqrt{(KT + D)\ln K}\big)$ regret bound conjectured by Cesa-Bianchi et al. [2019] in the case of variable, but bounded delays. Here, $K$ is the number of actions and $D$ is the total delay over $T$ rounds. We then introduce a new algorithm that lifts the requirement of bounded delays by using a wrapper that skips rounds with excessively large delays. The new algorithm maintains the same regret bound, but similar to its predecessor requires prior knowledge of $D$ and $T$. For this algorithm we then construct a novel doubling scheme that forgoes the prior knowledge requirement under the assumption that the delays are available at action time (rather than at loss observation time). This assumption is satisfied in a broad range of applications, including interaction with servers and service providers. The resulting oracle regret bound is of order $\min_\beta \big(|S_\beta| + \beta \ln K + (KT + D_\beta)/\beta\big)$, where $|S_\beta|$ is the number of observations with delay exceeding $\beta$, and $D_\beta$ is the total delay of observations with delay below $\beta$. The bound relaxes to $\mathcal{O}\big(\sqrt{(KT + D)\ln K}\big)$, but we also provide examples where $D_\beta \ll D$ and the oracle bound has a polynomially better dependence on the problem parameters.

## 1 Introduction

Multiarmed bandits is an algorithmic paradigm for sequential decision making with a growing range of industrial applications, including content recommendation, computational advertising, and many more. In the multiarmed bandit framework an algorithm repeatedly takes actions (e.g., recommendation of content to a user) and observes outcomes of these actions (e.g., whether the user engaged with the content), whereas the outcome of alternative actions (e.g., alternative content that could have been recommended) remains unobserved. In many real-life situations the algorithm experience delay between execution of an action and observation of its outcome. Within the delay period the algorithm may be forced to make a series of other actions (e.g., interact with new users) before observing the outcomes of all the previous actions. This setup falls outside of the classical multiarmed bandit paradigm, where observations happen instantaneously after the actions, and motivates the study of bandit algorithms that are provably robust in the presence of delays.

---

[*]Part of this work was done while visiting Università degli Studi di Milano, Milan, Italy

We focus on the nonstochastic (a.k.a. oblivious adversarial) bandit setting, where the losses faced by the algorithm are generated by an unspecified deterministic mechanism. Though it might be of adversarial intent, the mechanism is oblivious to internal randomization of the algorithm. In the delayed version, the loss of an action executed at time $t$ is observed at time $t+d_t$, where the delay $d_t$ is also chosen deterministically and obliviously. Thus, at time step $t$ the algorithm receives observations from time steps $s \le t$ for which $s + d_s = t$. This delay is the independent of the action chosen. The algorithm's performance is evaluated by regret, which is the difference between the algorithm's cumulative loss and the cumulative loss of the best static action in hindsight. The regret definition is the same as in the ordinary setting without delays. When all the delays are constant ($d_t = d$ for all $t$), the optimal regret is known to scale as $\mathcal{O}\big(\sqrt{(K+d)T \ln K}\big)$, where $T$ is the time horizon and $K$ is the number of actions [Cesa-Bianchi et al., 2019]. Remarkably, this bound is achieved by "delayed" `Exp3`, which is a minor modification of the standard `Exp3` algorithm performing updates as soon as the losses become available.

The case of variable delays has previously been studied in the full information setting by Joulani et al. [2016]. They prove a regret bound of order $\sqrt{(D+T)\ln K}$, where $D = \sum_{t=1}^{T} d_t$ is the total delay. Their proof is based on a generic reduction from delayed full information feedback to full information with no delay. The applicability of this technique to the bandit setting is unclear (see Appendix A). Cesa-Bianchi et al. [2019] conjecture an upper bound of order $\sqrt{(KT+D)\ln K}$ for the bandit setting with variable delays. Note that this bound cannot be improved in the general case because of the lower bound $\Omega\big(\sqrt{(K+d)T}\big)$, which holds for any $d$. In a recent paper, Li et al. [2019] study a harder variant of bandits, where the delays $d_t$ remain unknown. As a consequence, if an action is played at time $s$ and then more times in between time steps $s$ and $s + d_s$, the learner cannot tell which specific round the loss observed at time $s + d_s$ refers to. In this harder setting, for known $T$, $D$, and $d_{\max}$, Li et al. [2019] prove a regret bound of $\widetilde{\mathcal{O}}\big(\sqrt{d_{\max}K(T+D)}\big)$. Cesa-Bianchi et al. [2018] further study an even harder setting of bandits with anonymous composite feedback. In this setting at time step $t$ the learner observes feedback, which is a composition of partial losses of the actions taken in the last $d_{\max}$ rounds. In this setting Cesa-Bianchi et al. [2018] obtain an $\mathcal{O}\big(\sqrt{d_{\max}KT \ln K}\big)$ regret bound (which is tight within the $\ln K$ factor, and in fact tighter than the bound of Li et al. [2019] for an easier problem).

Our paper is structured in the following way. We start by investigating the regret of `Exp3` in the variable delay setting. We prove that for known $T$, $D$, and $d_{\max}$, and assuming that $d_{\max}$ is at most of order $\sqrt{(KT+D)/(\ln K)}$, "delayed" `Exp3` achieves the conjectured bound of $\mathcal{O}\big(\sqrt{(KT+D)\ln K}\big)$. In order to remove the restriction on $d_{\max}$ and eliminate the need of its knowledge we introduce a wrapper algorithm, `Skipper`. `Skipper` prevents the wrapped bandit algorithm from making updates using observations with delay exceeding a given threshold $\beta$. This threshold acts as a tunable upper bound on the delays observed by the underlying algorithm, so if $T$ and $D$ are known we can choose $\beta$ that attains the desired $\mathcal{O}\big(\sqrt{(KT+D)\ln K}\big)$ regret bound with "delayed" `Exp3` wrapped within `Skipper`.

To dispense of the need for knowing $T$ and $D$, the first approach coming to mind is the doubling trick. However, applying the standard doubling to $D$ is problematic, because the event that the actual total delay $d_1 + \cdots + d_t$ exceeds an estimate $D$ is observed at time $t + d_t$ rather than at time $t$. In order to address this issue, we consider a setting in which the algorithm observes the delay $d_t$ at time $t$ rather than at time $t + d_t$. To distinguish between this setting and the previous one we say that "the delay is observed at action time" if it is observed at time $t$ and "the delay is observed at observation time" if it is observed at time $t + d_t$. Observing the delay at action time is motivated by scenarios in which a learning agent depends on feedback from a third party, for instance a server or laboratory that processes the action in order to evaluate it. In such cases, the third party might partially control the delay, and provide the agent with a delay estimate based on contingent and possibly private information. In the server example the delay could depend on workload, while the laboratory might have processing times and an order backlog. Other examples include medical imaging where the availability of annotations depends on medical professionals work schedule. Common for these examples is that the third party knows the delay before the action is taken.

Within the "delay at action time" setting we achieve a much stronger regret bound. We show that `Skipper` wrapping delayed `Exp3` and combined with a carefully designed doubling trick enjoys an implicit regret bound of order $\min_\beta \big(|S_\beta| + \beta \ln K + (KT + D_\beta)/\beta\big)$, where $D_\beta$ is the total delay of observations with delay below $\beta$. This bound is attained without any assumptions on the sequence

Table 1: Spectrum of delayed feedback settings and the corresponding regret bounds, progressing from easier to harder settings. Results marked by (*) have matching lower bounds up to the $\sqrt{\ln K}$ factor. If all the delays are identical, then $D = dT$ and (**) has a lower bound following from Cesa-Bianchi et al. [2019] and matching up to the $\sqrt{\ln K}$ factor. However, for non-identical delays the regret can be much smaller, as we show in Example 8.

| Setting | Regret Bound | | Reference |
|---|---|---|---|
| Fixed delay | $\mathcal{O}\big(\sqrt{(K+d)T\ln K}\big)$ | (*) | Cesa-Bianchi et al. [2019] |
| Delay at action time | $\mathcal{O}\left(\min_\beta\left(\|S_\beta\| + \beta\ln K + \frac{KT+D_\beta}{\beta}\right)\right)$ | | This paper |
| Delay at observation time with known $T, D$ | $\mathcal{O}\big(\sqrt{(KT+D)\ln K}\big)$ | (**) | This paper |
| Anonymous, composite with known $d_{\max}$ | $\mathcal{O}\big(\sqrt{d_{\max}KT\ln K}\big)$ | (*) | Cesa-Bianchi et al. [2018] |

of delays $d_t$ and with no need for prior knowledge of $T$ and $D$. The implicit bound can be relaxed to an explicit bound of $\mathcal{O}\big(\sqrt{(KT+D)\ln K}\big)$, however if $D_\beta \ll D$ it can be much tighter. We provide an instance of such a problem in Example 8, where we get a polynomially tighter bound.

Table 1 summarizes the spectrum of delayed feedback models in the bandit case and places our results in the context of prior work.

## 1.1 Additional related work

Online learning with delays was pioneered by Mesterharm [2005] — see also [Mesterharm, 2007, Chapter 8]. More recent work in the full information setting include [Zinkevich et al., 2009, Quanrud and Khashabi, 2015, Ghosh and Ramchandran, 2018]. The theme of large or unbounded delays in the full information setting was also investigated by Mann et al. [2018] and Garrabrant et al. [2016]. Other related approaches are the works by Shamir and Szlak [2017], who use a semi-adversarial model, and Chapelle [2014], who studies the role of delays in the context of onlne advertising. Chapelle and Li [2011] perform an empirical study of the impact of delay in bandit models. This is extended in [Mandel et al., 2015]. The analysis of Exp3 in a delayed setting was initiated by Neu et al. [2014]. In the stochastic case, bandit learning with delayed feedback was studied in [Dudík et al., 2011, Vernade et al., 2017]. The results were extended to the anonymous setting by Pike-Burke et al. [2018] and by Garg and Akash [2019], and to the contextual setting by Arya and Yang [2019].

## 2 Setting and notation

We consider an oblivious adversarial multiarmed bandit setting, where $K$ sequences of losses are generated in an arbitrary way prior to the start of the game. The losses are denoted by $\ell_t^a$, where $t$ indexes the game rounds and $a \in \{1, \ldots, K\}$ indexes the sequences. We assume that all losses are in the $[0, 1]$ interval. We use the notation $[K] = \{1, \ldots, K\}$ for brevity. At each round of the game the learner picks an action $A_t$ and suffers the loss of that action. The loss $\ell_t^{A_t}$ is observed by the learner after $d_t$ rounds, where the sequence of delays $d_1, d_2, \ldots$ is determined in an arbitrary way before the game starts. Thus, at round $t$ the learner observes the losses of prior actions $A_s$ for which $s + d_s = t$. We assume that the losses are observed "at the end of round $t$", *after* the action $A_t$ has been selected. We consider two different settings for receiving information about the delays $d_t$:

**Delay available at observation time** The delay $d_t$ is observed when the feedback $\ell_t^{A_t}$ arrives at the end of round $t + d_t$. This corresponds to the feedback being timestamped.

**Delay available at action time** The delay $d_t$ is observed at the beginning of round $t$, prior to selecting the action $A_t$.

The following learning protocol provides a formal description of our setting.

> **Protocol for bandits with delayed feedback**
> For $t = 1, 2, \ldots$
>  1. If *delay is available at action time*, then $d_t \geq 0$ is revealed to the learner
>  2. The learner picks an action $A_t \in \{1, \ldots, K\}$ and suffers the loss $\ell_t^{A_t} \in [0, 1]$
>  3. Pairs $\left(s, \ell_s^{A_s}\right)$ for all $s \leq t$ such that $s + d_s = t$ are observed

We measure the performance of the learner by her *expected regret* $\bar{\mathcal{R}}_T$, which is defined as the difference between the expected cumulative loss of the learner and the loss of the best static strategy in hindsight:

$$\bar{\mathcal{R}}_T = \mathbb{E}\left[\sum_{t=1}^{T} \ell_t^{A_t}\right] - \min_a \sum_{t=1}^{T} \ell_t^a.$$

This regret definition is the same as the one used in the standard multiarmed bandit setting without delay.

## 3   Delay available at observation time: Algorithms and results

This section deals with the first of our two settings, namely when delays are observed together with the losses. We first introduce a modified version of "delayed" Exp3, which we name Delayed Exponential Weights (DEW) and which is capable of handling variable delays. We then introduce a wrapper algorithm, `Skipper`, which filters out excessively large delays. The two algorithms also serve as the basis for the next section, where we provide yet another wrapper for tuning the parameters of `Skipper`.

### 3.1   Delayed Exponential Weights (DEW)

`DEW` is an extension of the standard exponential weights approach to handle delayed feedback. The algorithm, laid out in Algorithm 1, performs an exponential update using every individual feedback as it arrives, which means that between each prediction either zero, one, or multiple updates might occur. The algorithm assumes that the delays are bounded and that an upper bound $d_{\max} \geq \max_t d_t$ on the delays is known.

---
**Algorithm 1:** Delayed exponential weights (DEW)

---
**Input :** Learning rate $\eta$; upper bound on the delays $d_{\max}$

Truncate the learning rate: $\eta' = \min\{\eta, (4ed_{\max})^{-1}\}$;

Initialize $w_0^a = 1$ for all $a \in [K]$;
**for** $t = 1, 2, \ldots$ **do**
> Let $p_t^a = \frac{w_{t-1}^a}{\sum_b w_{t-1}^b}$ for $a \in [K]$;
> Draw an action $A_t \in [K]$ according to the distribution $\boldsymbol{p_t}$ and play it;
> Observe feedback $(s, \ell_s^{A_s})$ for all $\{s : s + d_s = t\}$ and construct estimators $\hat{\ell}_s^a = \frac{\ell_s^a \mathbb{1}(a = A_s)}{p_s^a}$;
> Update $w_t^a = w_{t-1}^a \exp\left(-\eta' \sum_{s:s+d_s=t} \hat{\ell}_s^a\right)$;

**end**

---

The following theorem provides a regret bound for Algorithm 1. The bound is a generalization of a similar bound in Cesa-Bianchi et al. [2019].

**Theorem 1.** *Under the assumption that an upper bound on the delays $d_{\max}$ is known, the regret of Algorithm 1 with a learning rate $\eta$ against an oblivious adversary satisfies*

$$\bar{\mathcal{R}}_T \leq \max\left\{\frac{\ln K}{\eta}, 4ed_{\max}\ln K\right\} + \eta\left(\frac{KTe}{2} + D\right),$$

where $D = \sum_{t=1}^{T} d_t$. In particular, if $T$ and $D$ are known and $\eta = \sqrt{\frac{\ln K}{\frac{KTe}{2} + D}} \leq \frac{1}{4ed_{\max}}$, we have

$$\bar{\mathcal{R}}_T \leq 2\sqrt{\left(\frac{KTe}{2} + D\right)\ln K}. \tag{1}$$

The proof of Theorem 1 is based on proving the stability of the algorithm across rounds. The proof is sketched out in Section 5. As Theorem 1 shows, Algorithm 1 performs well if $d_{\max}$ is small and we also have preliminary knowledge of $d_{\max}$, $T$, and $D$. However, a single delay of order $T$ increases $d_{\max}$ up to order $T$, which leads to a linear regret bound in Theorem 1. This is an undesired property, which we address with the skipping scheme presented next.

### 3.2 Skipping scheme

We introduce a wrapper for Algorithm 1, called `Skipper`, which disregards feedback from rounds with excessively large delays. The regret in the skipped rounds is trivially bounded by 1 (because the losses are assumed to be in $[0, 1]$) and the rounds are taken out of the analysis of the regret of `DEW`. `Skipper` operates with an externally provided threshold $\beta$ and skips all rounds where $d_t \geq \beta$. The advantage of skipping is that it provides a natural upper bound on the delays for the subset of rounds processed by `DEW`, $d_{\max} = \beta$. Thus, we eliminate the need of knowledge of the maximal delay in the original problem. The cost of skipping is the number of skipped rounds, denoted by $|S_\beta|$, as captured in Lemma 2. Below we provide a regret bound for the combination of `Skipper` and `DEW`.

---

**Algorithm 2:** Skipper

**Input :** Threshold $\beta$; Algorithm $\mathcal{A}$.

**for** $t = 1, 2, \ldots$ **do**

    Get prediction $A_t$ from $\mathcal{A}$ and play it;

    Observe feedback $(s, \ell_s^{A_s})$ for all $\{s : s + d_s = t\}$, and feed it to $\mathcal{A}$ for each $s$ with $d_s < \beta$;

**end**

---

**Lemma 2.** *The expected regret of* `Skipper` *with base algorithm $\mathcal{A}$ and threshold parameter $\beta$ satisfies*

$$\bar{\mathcal{R}}_T \leq |S_\beta| + \bar{\mathcal{R}}_{T \backslash S_\beta}, \tag{2}$$

*where $|S_\beta|$ is the number of skipped rounds (those for which $d_t \geq \beta$) and $\bar{\mathcal{R}}_{T \backslash S_\beta}$ is a regret bound for running $\mathcal{A}$ on the subset of rounds $[T] \backslash S_\beta$ (those, for which $d_t < \beta$).*

A proof of the lemma is found in Appendix C. When combined with the previous analysis for `DEW`, Lemma 2 gives us the following regret bound.

**Theorem 3.** *The expected regret of* `Skipper`$(\beta, \text{DEW}(\eta, \beta))$ *against an oblivious adversary satisfies*

$$\bar{\mathcal{R}}_T \leq |S_\beta| + \max\left\{\frac{\ln K}{\eta}, 4e\beta\ln K\right\} + \eta\left(\frac{KTe}{2} + D_\beta\right), \tag{3}$$

*where $D_\beta = \sum_{t \notin S_\beta} d_t$ is the cumulative delay experienced by* `DEW`.

*Proof.* Theorem 1 holds for parameters $(\eta, \beta)$ for `DEW` run under `Skipper`. We then apply Lemma 2. $\square$

**Corollary 4.** *Assume that $T$ and $D$ are known and take*

$$\eta = \frac{1}{4e\beta}, \quad \beta = \sqrt{\frac{\frac{eKT/2+D}{4e} + D}{4e\ln K}}.$$

*Then the expected regret of* `Skipper`$(\beta, \text{DEW}(\eta, \beta))$ *against an oblivious adversary satisfies*

$$\bar{\mathcal{R}}_T \leq 2\sqrt{\left(\frac{KTe}{2} + (1 + 4e)D\right)\ln K}.$$

*Proof.* Note that $D \geq \beta |S_\beta| \Rightarrow |S_\beta| \leq \frac{D}{\beta}$. By substituting this into (3), observing that $D_\beta \leq D$, and substituting the values of $\eta$ and $\beta$ we obtain the result. $\qquad\square$

Note that Corollary 4 recovers the regret scaling in Theorem 1, equation (1) within constant factors in front of $D$ without the need of knowledge of $d_{\max}$. Similar to Theorem 1, Corollary 4 is tight in the worst case. The tuning of $\beta$ still requires the knowledge of $T$ and $D$. In the next section we get rid of this requirement.

## 4   Delay available at action time: Oracle tuning and results

This section deals with the second setting, where the delays are observed before taking an action. The combined algorithm introduced in the previous section relies on prior knowledge of $T$ and $D$ for tuning the parameters. In this section we eliminate this requirement by leveraging the added information about the delays at the time of action. The information is used in an implicit doubling scheme for tuning `Skipper`'s threshold parameter $\beta$. Additionally, the new bound scales with the experienced delay $D_\beta$ rather than the full delay $D$ and is significantly tighter when $D_\beta \ll D$. This is achieved through direct optimization of the regret bound in terms of $|S_\beta|$ and $D_\beta$, as opposed to Corollary 4, which tunes $\beta$ using the potentially loose inequality $|S_\beta| \leq D/\beta$.

### 4.1   Setup

Let $m$ index the *epochs* of the doubling scheme. In each epoch we restart the algorithm with new parameters and continually monitor the termination condition in equation (6). The learning rate within epoch $m$ is set to $\eta_m = \frac{1}{4e\beta_m}$, where $\beta_m$ is the threshold parameter of the epoch. Theorem 3 provides a regret bound for epoch $m$ denoted by

$$\text{Bound}_m(\beta_m) := |S_{\beta_m}^m| + 4e\beta_m \ln K + \frac{\sigma(m)eK/2 + D_{\beta_m}^m}{4e\beta_m}, \qquad (4)$$

where $\sigma(m)$ denotes the length of epoch $m$ and $|S_{\beta_m}^m|$ and $D_{\beta_m}^m$ are, respectively, the number of skipped rounds and the experienced delay within epoch $m$.

Let $\omega_m = 2^m$. In epoch $m$ we set

$$\beta_m = \frac{\sqrt{\omega_m}}{4e \ln K} \qquad (5)$$

and we stay in epoch $m$ as long as the following condition holds:

$$\max\left\{ |S_{\beta_m}^m|^2, \left( \frac{eK\sigma(m)}{2} + D_{\beta_m}^m \right) \ln K \right\} \leq \omega_m. \qquad (6)$$

Since $d_t$ is observed at the beginning of round $t$, we are able to evaluate condition (6) and start a new epoch before making the selection of $A_t$. This provides the desired tuning of $\beta_m$ for all rounds without the need of a separate treatment of epoch transition points.

While being more elaborate, this doubling scheme maintains the intuition of standard approaches. First of all, the condition for doubling (6) ensures that the regret bound in each period is optimized by explicitly balancing the contribution of each term in equation (4). Secondly, the geometric progression of the tuning (5) —and thus of the resulting regret bounds— means that the total regret bound summed over the epochs can be bounded in relation to the bound in the final completed epoch.

In the following we refer to the doubling scheme defined by (5) and (6) as `Doubling`.

### 4.2   Results

The following results show that the proposed doubling scheme works as well as oracle tuning of $\beta$ when the learning rate is fixed at $\eta = 1/4e\beta$. We first compare our performance to the optimal tuning in a single epoch, where we let

$$\beta_m^* = \arg\min_{\beta_m} \text{Bound}_m(\beta_m) \qquad (7)$$

be the minimizer of (4).

**Lemma 5.** *The regret bound* (4) *for any non-final epoch* $m$*, with the epochs and* $\beta_m$ *controlled by* `Doubling` *satisfies*

$$\text{Bound}_m(\beta_m) \leq 3\sqrt{\omega_m} \leq 3\,\text{Bound}_m(\beta_m^*) + 2e^2 K \ln K + 1. \tag{8}$$

The lemma is the main machinery of the analysis of `Doubling` and its proof is provided in Appendix C. Applying it to `Skipper`$(\beta, \text{DEW}(\eta, \beta))$ leads to the following main result.

**Theorem 6.** *The expected regret of* `Skipper`$(\beta, \text{DEW}(\eta, \beta))$ *tuned by* `Doubling` *satisfies for any* $T$

$$\bar{\mathcal{R}}_T \leq 15 \min_\beta \left\{ |S_\beta| + 4e\beta \ln K + \frac{KT + D_\beta}{4e\beta} \right\} + 10e^2 K \ln K + 5.$$

The proof of Theorem 6 is based on Lemma 5 and is provided in Appendix C.

**Corollary 7.** *The expected regret of* `Skipper`$(\beta, \text{DEW}(\eta, \beta))$ *tuned by* `Doubling` *can be relaxed for any* $T$ *to*

$$\bar{\mathcal{R}}_T \leq 30\sqrt{\left( \frac{KTe}{2} + (1 + 4e)D \right) \ln K} + 10e^2 K \ln K + 5. \tag{9}$$

*Proof.* The first term in the bound of Theorem 6 can be directly bounded using Corollary 4. $\qquad\square$

Note that both Theorem 6 and Corollary 7 require no knowledge of $T$ and $D$.

### 4.3 Comparison of the oracle and explicit bounds

We finish the section with a comparison of the oracle bound in Theorem 6 and the explicit bound in Corollary 7. Ignoring the constant and additive terms, the bounds are

$$\text{explicit} \quad : \quad \mathcal{O}\left( \sqrt{(KT + D) \ln K} \right),$$

$$\text{oracle} \quad : \quad \mathcal{O}\left( \min_\beta \left\{ |S_\beta| + \beta \ln K + \frac{KT + D_\beta}{\beta} \right\} \right).$$

Note that the oracle bound is always as strong as the explicit bound. There are, however, cases where it is much tighter. Consider the following example.

**Example 8.** *For* $t < \sqrt{KT / \ln K}$ *let* $d_t = T - t$ *and for* $t \geq \sqrt{KT / \ln K}$ *let* $d_t = 0$*. Take* $\beta = \sqrt{KT / \ln K}$*. Then* $D = \Theta(T\sqrt{KT / \ln K})$*, but* $D_\beta = 0$ *(assuming that* $T \geq K \ln K$*) and* $|S_\beta| < \sqrt{KT / \ln K}$*. The corresponding regret bounds are*

$$\text{explicit} \quad : \quad \mathcal{O}\left( \sqrt{KT \ln K + T\sqrt{KT}} \right) = \mathcal{O}(T^{3/4}),$$

$$\text{oracle} \quad : \quad \mathcal{O}\left( \sqrt{KT \ln K} \right) = \mathcal{O}(T^{1/2}).$$

## 5 Analysis of Algorithm 1

This section contains the main points of the analysis of Algorithm 1 leading to the proof of Theorem 1 which were postponed from Section 3. Full proofs are found in Appendix B.

The analysis is a generalization of the analysis of delayed `Exp3` in Cesa-Bianchi et al. [2019], and consists of a general regret analysis and two stability lemmas.

### 5.1 Additional notation

We let $N_t = |\{s : s + d_s \in [t, t + d_t)\}|$ denote the *stability-span* of $t$, which is the amount of feedback that arrives between playing action $A_t$ and observing its feedback. Note that letting $N = \max_t N_t$ we have $N \leq 2\max_t d_t \leq 2d_{\max}$, since this may include feedback from up to $\max_s d_s$ rounds prior to round $t$ and up to $d_t$ rounds after round $t$.

We introduce $\mathcal{Z} = (z_1, ..., z_T)$ to be a permutation of $[T] = \{1, ..., T\}$ sorted in ascending order according to the value of $z + d_z$ with ties broken randomly, and let $\Psi_i = (z_1, ..., z_i)$ be its first $i$ elements. Similarly, we also introduce $\mathcal{Z}'_t = (z'_1, ..., z'_{N_t})$ as an enumeration of $\{s : s + d_s \in [t, t + d_t)\}$.

For a subset the integers $C$, corresponding to timesteps, we also introduce

$$q^a(C) = \frac{\exp\left(-\eta' \sum_{s \in C} \hat{\ell}^a_s\right)}{\sum_b \exp\left(-\eta' \sum_{s \in C} \hat{\ell}^b_s\right)}. \tag{10}$$

The nominator and denominator in the above expression will also be denoted by $w^a(C)$ and $W(C)$ corresponding to the definition of $p^a_t$.

By finally letting $C_{t-1} = \{s : s + d_s < t\}$ we have $p^a_t = q^a(C_{t-1})$.

## 5.2 Analysis of delayed exponential weights

The starting point is the following modification of the basic lemma within the Exp3 analysis that takes care of delayed updates of the weights.

**Lemma 9.** *Algorithm 1 satisfies*

$$\sum_{t=1}^{T} \sum_{a=1}^{K} p^a_{t+d_t} \hat{\ell}^a_t - \min_{a \in [K]} \sum_{t} \hat{\ell}^a_t \leq \frac{\ln K}{\eta'} + \frac{\eta}{2} \sum_{t=1}^{T} \sum_{a=1}^{K} p^a_{t+d_t} \left(\hat{\ell}^a_t\right)^2. \tag{11}$$

To make use of Lemma 9, we need to figure out the relationship between $p^a_{t+d_t}$ and $p^a_t$. This is achieved by the following two lemmas, which are generalizations and refinements of Lemmas 1 and 2 in Cesa-Bianchi et al. [2019].

**Lemma 10.** *When using Algorithm 1 the resulting probabilities fulfil for every $t$ and $a$*

$$p^a_{t+d_t} - p^a_t \geq -\eta' \sum_{i=1}^{N_t} q^a\left(C_{t-1} \cup \{z'_j : j < i\}\right) \hat{\ell}^a_{z'_i}, \tag{12}$$

*where $z'_j$ is an enumeration of $\{s : s + d_s \in [t, t + d_t)\}$.*

The above lemma allows us to bound $p^a_{t+d_t}$ from below in terms of $p^a_t$. We similarly need to be able to upper bound the probability, which is captured in the second probability drift lemma.

**Lemma 11.** *The probabilities defined by* (10) *satisfy for any $i$*

$$q^a(\Psi_i) \leq \left(1 + \frac{1}{2N - 1}\right) q^a(\Psi_{i-1}). \tag{13}$$

## 5.3 Proof sketch of Theorem 1

By using Lemma 10 to bound the left hand side of (11) we have

$$\sum_t \sum_a p^a_t \hat{\ell}^a_t - \min_a \sum_t \hat{\ell}^a_t \leq \frac{\ln K}{\eta'} + \frac{\eta'}{2} \sum_{t=1}^{T} \sum_{a=1}^{K} p^a_{t+d_t} \left(\hat{\ell}^a_t\right)^2$$

$$+ \eta' \sum_t \sum_a \hat{\ell}^a_t \sum_{i=1}^{N_t} q^a\left(C_{t-1} \cup \{z'_j : j < i\}\right) \hat{\ell}^a_{z'_i}.$$

Repeated use of Lemma 11 bounds the second term on the right hand side by $\eta' TKe/2$ in expectation. The third term on the right hand side can be bounded by $D$. Taking the maximum over the two possible values of the truncated learning rate finishes the proof. □

# 6 Discussion

We have presented an algorithm for multiarmed bandits with variably delayed feedback, which achieves the $\mathcal{O}\big(\sqrt{(KT+D)\ln K}\big)$ regret bound conjectured by Cesa-Bianchi et al. [2019]. The algorithm is based on a procedure for skipping rounds with excessively large delays and refined analysis of the exponential weights algorithm with delayed observations. At the moment the skipping procedure requires prior knowledge of $T$ and $D$ for tuning the skipping threshold. However, if the delay information is available "at action time", as in the examples described in the introduction, we provide a sophisticated doubling scheme for tuning the skipping threshold that requires no prior knowledge of $T$ and $D$. Furthermore, the refined tuning also leads to a refined regret bound of order $\mathcal{O}\big(\min_\beta\big(|S_\beta| + \beta\ln K + \frac{KT+D_\beta}{\beta}\big)\big)$, which is polynomially tighter when $D_\beta \ll D$. We provide an example of such a problem in the paper.

Our work leads to a number of interesting research questions. The main one is whether the two regret bounds are achievable when the delays are available "at observation time" without prior knowledge of $D$ and $T$. Alternatively, is it possible to derive lower bounds demonstrating the impossibility of further relaxation of the assumptions? More generally, it would be interesting to have refined lower bounds for problems with variably delayed feedback. Another interesting direction is a design of anytime algorithms, which do not rely on the doubling trick. Such algorithms can be used, for example, for achieving simultaneous optimality in stochastic and adversarial setups [Zimmert and Seldin, 2019a]. While a variety of anytime algorithms is available for non-delayed bandits, the extension to delayed feedback does not seem trivial. Some of these questions are addressed in a follow-up work by Zimmert and Seldin [2019b].

### Acknowledgments

Nicolò Cesa-Bianchi gratefully acknowledges partial support by the Google Focused Award *Algorithms and Learning for AI* (ALL4AI) and by the MIUR PRIN grant *Algorithms, Games, and Digital Markets* (ALGADIMAR). Yevgeny Seldin acknowledges partial support by the Independent Research Fund Denmark, grant number 9040-00361B.

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
