[Supplementary Material · Unrestricted_Delays_2_supplementary.pdf]

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

# A  Alternative approaches

This appendix is an addition to the discussion of relevant literature in the introduction.

The present paper follows an approach to delayed feedback based on explicitly analysing exponential weights with delays and considering the stability of this class of algorithms. An alternative approach in literature is instead to construct a reduction from the delayed case to the undelayed case and thus circumventing the need for direct analysis of the underlying algorithm, since this will usually be well established. Such a reduction is done in the full information case by Joulani et al. [2016] but with no mention of how it might apply to the bandit case. Below we briefly sketch their reduction in the case of OCO with linear loss functions, which specializes to bandits.

Let $t$ index time and $s$ index *virtual rounds*, meaning rounds where an update is made. In other words in every round $t$ the algorithm makes a prediction, while in every round $s$ the algorithm receives *one* point of feedback. Quantities indexed by virtual rounds are further denoted by a tilde. $\tilde{\tau}_s$ is the number of virtual rounds (equivalently updates) between when the action giving rise to loss $\tilde{\ell}_s$ is played and the loss is received. Let the function $\rho$ map a virtual round $s$ where $\tilde{\ell}_s$ is observed to the round $t = \rho(s)$ where it is played. As such $\ell_{\rho(s)} = \tilde{\ell}_s$, but $p_{\rho(s)} = \tilde{p}_{s-\tilde{\tau}_s}$.

**Deterministic case:**  In the full information case for deterministic losses, the actions (probability distributions) played by the algorithm does not depend on any randomness, since the feedback is not dependent on the action played. The expected regret can thus be regarded as deterministic, and the following reduction can be carried out:

$$
\begin{aligned}
\mathcal{R}_T &= \sum_t \left[ \sum_a \ell_t^a p_t^a - \ell_t^\star \right] \\
&= \sum_s \left[ \sum_a \ell_{\rho(s)}^a p_{\rho(s)}^a - \ell_{\rho(s)}^\star \right] \\
&= \sum_s \left[ \sum_a \tilde{\ell}_s^a \tilde{p}_{s-\tilde{\tau}_s}^a - \tilde{\ell}_s^\star \right] \\
&= \sum_s \sum_a \tilde{\ell}_s^a \left( \tilde{p}_{s-\tilde{\tau}_s}^a - \tilde{p}_s^a \right) + \sum_s \left[ \sum_a \tilde{\ell}_s^a \tilde{p}_s^a - \tilde{\ell}_s^\star \right],
\end{aligned}
$$

where we let $\star$ denote the optimal action in hindsight. The point of this calculation is that the final term above is the regret of the undelayed base algorithm, while the first term is an additive drift term, similar to what we are considering in Lemma 10.

**Conditional case:**  To extend this to bandits, we need to consider the case where the actions (probability distributions) of the algorithm depends on the internal randomness. The expected regret then requires taking expectation over this randomness:

$$
\begin{aligned}
\bar{\mathcal{R}}_T &= \mathop{\mathbb{E}}_{A_1,\ldots,A_T} \left[ \sum_t \left[ \sum_a \ell_t^a p_t^a - \ell_t^\star \right] \right] \\
&= \mathop{\mathbb{E}}_{A_1,\ldots,A_T} \left[ \sum_s \sum_a \tilde{\ell}_s^a \left( \tilde{p}_{s-\tilde{\tau}_s}^a - \tilde{p}_s^a \right) \right] + \mathop{\mathbb{E}}_{A_1,\ldots,A_T} \left[ \sum_s \left[ \sum_a \tilde{\ell}_s^a \tilde{p}_s^a - \tilde{\ell}_s^\star \right] \right].
\end{aligned}
$$

Now however the final term is no longer the expected regret of the underlying algorithm without delays, since the conditional expectations taken here are not the same as they would be for the undelayed algorithm. In particular the order of the conditional expectations might be different since the delays are not the same, so the reduction is not directly applicable.

# B  Full proof of Theorem 1

This appendix contains the full analysis of Algorithm 1, i.e., proofs of the lemmas in Section 5 and the full proof of Theorem 1.

## B.1 Proof of Lemma 9

We consider the quantity

$$
\frac{W_t}{W_{t-1}} = \frac{\sum_a w_{t-1}^a \prod_{s:s+d_s=t} \exp\left(-\eta' \hat{\ell}_s^a\right)}{W_{t-1}}
$$

$$
= \sum_a p_t^a \prod_{s:s+d_s=t} \exp\left(-\eta' \hat{\ell}_s^a\right)
$$

$$
\leq \sum_a p_t^a \sum_{s:s+d_s=t} \exp\left(-\eta' \hat{\ell}_s^a\right)
$$

$$
\leq \sum_a p_t^a \sum_{s:s+d_s=t} \left(1 - \eta' \hat{\ell}_s^a + \frac{\eta'^2}{2}\left(\hat{\ell}_s^a\right)^2\right)
$$

$$
= \sum_{s:s+d_s=t} \left(1 - \eta' \sum_a p_t^a \hat{\ell}_s^a + \frac{\eta'^2}{2}\sum_a p_t^a \left(\hat{\ell}_s^a\right)^2\right)
$$

$$
= 1 + |\{s : s+d_s=t\}| - 1 - \eta' \sum_{s:s+d_s=t}\sum_a p_t^a \hat{\ell}_s^a + \frac{\eta'^2}{2}\sum_{s:s+d_s=t}\sum_a p_t^a \left(\hat{\ell}_s^a\right)^2
$$

$$
\leq \exp\left(|\{s : s+d_s=t\}| - 1 - \eta' \sum_{s:s+d_s=t}\sum_a p_t^a \hat{\ell}_s^a + \frac{\eta'^2}{2}\sum_{s:s+d_s=t}\sum_a p_t^a \left(\hat{\ell}_s^a\right)^2\right),
$$

where the first inequality uses that each $\exp\left(-\eta' \hat{\ell}_s^a\right)$ is in $(0,1]$, the second inequality uses $e^x \leq 1 + x + x^2/2$ for $x \leq 0$, and the final inequality uses $e^x \geq 1 + x$ for all $x$.

By a telescoping sum and the above we get

$$
\frac{W_T}{W_0} \leq \exp\left(-\eta' \sum_t \sum_{s:s+d_s=t}\sum_a p_t^a \hat{\ell}_s^a + \frac{\eta'^2}{2}\sum_t \sum_{s:s+d_s=t}\sum_a p_t^a \left(\hat{\ell}_s^a\right)^2\right), \qquad (14)
$$

using that $\sum_{t=1}^T |\{s : s+d_s=t\}| \leq T$. We also lower bound this fraction as

$$
\frac{W_T}{W_0} \geq \frac{\max_a \exp\left(-\eta' \sum_{s:s+d_s \leq T} \hat{\ell}_s^a\right)}{K}
$$

$$
\geq \frac{\max_a \exp\left(-\eta' \sum_{s=1}^T \hat{\ell}_s^a\right)}{K}
$$

$$
\geq \frac{\exp\left(-\eta' \min_a \sum_{s=1}^T \hat{\ell}_s^a\right)}{K}. \qquad (15)
$$

The proof is completed by combining (14) and (15), taking logarithms and rearranging, and noting that the sums of the form $\sum_t \sum_{s:s+d_s=t}$ only include each value of $s$ once, and thus are equivalent to summing over $s$ and identifying $t = s + d_s$. □

## B.2 Proof of Lemma 10

Note for any set of integers $C$ containing a value $x$, we have

$$
W(C) = \sum_a e^{-\eta' \hat{\ell}_x^a} e^{-\eta' \sum_{s \in C \setminus \{x\}} \hat{\ell}_s^a} \leq \sum_a e^{-\eta' \sum_{s \in C \setminus \{x\}} \hat{\ell}_s^a} = W(C \setminus \{x\}),
$$

which means

$$
q^a(C) = \frac{w^a(C)}{W(C)} \geq \frac{w^a(C)}{W(C \setminus \{x\})} = e^{-\eta' \hat{\ell}_x^a}\frac{w^a(C \setminus \{x\})}{W(C \setminus \{x\})} = e^{-\eta' \hat{\ell}_x^a} q^a(C \setminus \{x\}).
$$

This in turn implies

$$q^a(C) - q^a(C \backslash \{x\}) \geq \left( e^{-\eta' \hat{\ell}_x^a} - 1 \right) q^a(C \backslash \{x\}) \geq -\eta' \hat{\ell}_x^a q^a(C \backslash \{x\}).$$

Telescoping this over the individual observations $z_1', ..., z_{N_t}'$ we get

$$
\begin{aligned}
p_{t+d_t}^a - p_t^a &= q^a(C_{t+d_t-1}) - q^a(C_{t-1}) \\
&= \sum_{i=1}^{N_t} q^a \left( C_{t-1} \cup \{z_j' : j \leq i\} \right) - q^a \left( C_{t-1} \cup \{z_j' : j < i\} \right) \\
&\geq -\eta' \sum_{i=1}^{N_t} \hat{\ell}_{z_i'}^a q^a \left( C_{t-1} \cup \{z_j' : j < i\} \right)
\end{aligned}
$$

### B.3 Proof of Lemma 11

We prove the lemma by induction. For the base case, consider $i = 1$, where $\Psi_0 = \emptyset$, and thus $q^a(\Psi_{i-1}) = q^a(\Psi_0) = 1/K$. The maximal increase of $q^a$ by making a single observation will be if another arm is chosen and receives a loss of 1, making the loss estimator equal to $K$. This means

$$q^a(\Psi_i) \leq \frac{1}{K-1+e^{-\eta' K}} \leq \frac{1}{K - \eta' K} \leq \frac{1/K}{1 - \frac{1}{e2N}} \leq \frac{1/K}{1 - \frac{1}{2N}} = \frac{1}{K} \left( 1 + \frac{1}{2N-1} \right),$$

where first use $e^x \geq 1 + x$ for all $x$ and secondly use the upper bound on $\eta'$. since $1/K = q^a(\Psi_0)$ the base case is shown.

For the general case, assume that the lemma holds for $i - 1$. We first show that

$$
\begin{aligned}
q^a(\Psi_{i-1}) &\geq e^{-\eta' \hat{\ell}_{z_i}^a} q^a(\Psi_{i-1}) \\
&= \frac{w^a(\Psi_i)}{W(\Psi_{i-1})} \\
&= \frac{q^a(\Psi_i) W(\Psi_i)}{W(\Psi_{i-1})} \\
&= q^a(\Psi_i) \sum_b \frac{e^{-\eta' \hat{\ell}_{z_i}^b} w^b(\Psi_{i-1})}{W(\Psi_{i-1})} \\
&= q^a(\Psi_i) \sum_b e^{-\eta' \hat{\ell}_{z_i}^b} q^b(\Psi_{i-1}) \\
&\geq q^a(\Psi_i) \left( 1 - \eta' \sum_b \hat{\ell}_{z_i}^b q^b(\Psi_{i-1}) \right).
\end{aligned}
\tag{16}
$$

By expanding the loss estimator we get

$$\sum_b \hat{\ell}_{z_i}^b q^b(\Psi_{i-1}) = \hat{\ell}_{z_i}^{A_{z_i}} q^{A_{z_i}}(\Psi_{i-1}) \leq \frac{q^{A_{z_i}}(\Psi_{i-1})}{q^{A_{z_i}}(C_{z_i-1})} = \frac{q^{A_{z_i}}(\Psi_{i-1})}{q^{A_{z_i}}(\Psi_{l(i)})}, \tag{17}$$

by using $\ell_{z_i}^b \mathbb{1}(A_{z_i} = b) \leq 1$ for $b = A_{z_i}$ and the loss estimator is identically zero for all other $b$. We here define $l(i)$ as the index in $\mathcal{Z}$ of the last observation before round $z_i$. We now consider the difference in these indices, namely $(i - 1) - l(i)$.

Note that the loss from $z_i$ is observed at time $z_i + d_{z_i}$, but the losses from rounds $z_{i-1}, z_{i-2}, ...$ could potentially also be observed at this point. This means that all observations of losses from rounds $\Psi_{i-1} \backslash \Psi_{l(i)}$ are found in $[z_i, z_i + d_{z_i}]$. As maximally $N$ observations can be made both in $[z_i, z_i + d_{z_i})$ and in $[z_i + d_{z_i}, z_i + d_{z_i}]$ by assumption, and these $2N$ observations must include the observation of the loss from round $z_i$, we have a bound of

$$(i - 1) - l(i) \leq 2N - 1. \tag{18}$$

Telescoping the probability ratio and using the inductive assumption, we thus have

$$\frac{q^{A_{z_i}}(\Psi_{i-1})}{q^{A_{z_i}}(\Psi_{l(i)})} = \prod_{j=l(i)+1}^{i-1} \frac{q^{A_{z_i}}(\Psi_j)}{q^{A_{z_i}}(\Psi_{j-1})}$$

$$\leq \prod_{j=l(i)+1}^{i-1} \left(1 + \frac{1}{2N-1}\right)$$

$$= \left(1 + \frac{1}{2N-1}\right)^{2N-1}$$

$$\leq e. \tag{19}$$

Inserting this into (16) and using the upper bound on the learning rate gives us

$$q^a(\Psi_{i-1}) \geq q^a(\Psi_i)(1 - \eta' e)$$

$$\geq q^a(\Psi_i)\left(1 - \frac{1}{2N}\right),$$

which rearranges to the lemma statement. This concludes the inductive step $\qquad\square$.

## B.4   Full proof of Theorem 1

We start by combining Lemmas 9, 10 and 11 in the following way. By using Lemma 10 to bound the left hand side of (11) we have

$$\sum_t \sum_a p_{t+d_t}^a \hat{\ell}_t^a \geq \sum_t \sum_a p_t^a \hat{\ell}_t^a - \eta' \sum_t \sum_a \hat{\ell}_t^a \sum_{i=1}^{N_t} q^a \left(C_{t-1} \cup \{z_j' : j < i\}\right) \hat{\ell}_{z_i'}^a, \tag{20}$$

subtracting the final term gives us:

$$\sum_t \sum_a p_t^a \hat{\ell}_t^a - \min_a \sum_t \hat{\ell}_t^a \leq \frac{\ln K}{\eta'} + \frac{\eta'}{2} \sum_{t=1}^{T} \sum_{a=1}^{K} p_{t+d_t}^a \left(\hat{\ell}_t^a\right)^2$$

$$+ \eta' \sum_t \sum_a \hat{\ell}_t^a \sum_{i=1}^{N_t} q^a \left(C_{t-1} \cup \{z_j' : j < i\}\right) \hat{\ell}_{z_i'}^a. \tag{21}$$

Where we note that the left hand side becomes the expected regret when taking expectations over the choice of $A_t$.

The second term on the right hand side of (21) can be bounded by repeated use of Lemma 11:

$$\sum_t \sum_a p_{t+d_t}^a \left(\hat{\ell}_t^a\right)^2 = \sum_t \sum_a p_t^a \frac{p_{t+d_t}^a}{p_t^a} \left(\hat{\ell}_t^a\right)^2$$

$$= \sum_t \sum_a p_t^a \left(\hat{\ell}_t^a\right)^2 \prod_{i=1}^{N_t} \frac{q^a(C_{t-1} \cup \{z_j' : j \leq i\})}{q^a(C_{t-1} \cup \{z_j' : j < i\})}$$

$$\leq \sum_t \sum_a p_t^a \left(\hat{\ell}_t^a\right)^2 \left(1 + \frac{1}{2N-1}\right)^{N_t}$$

$$\leq \sum_t \sum_a p_t^a \left(\hat{\ell}_t^a\right)^2 \left(1 + \frac{1}{2N-1}\right)^{2N-1}$$

$$\leq \sum_t \sum_a p_t^a \left(\hat{\ell}_t^a\right)^2 e,$$

which in expectation is bounded by $TKe$.

The final term in (21) requires a bit more work. We first note that:

$$\mathbb{E}\left[\sum_t \sum_a \hat{\ell}_t^a \sum_{i=1}^{N_t} q^a \left(C_{t-1} \cup \{z_j' : j < i\}\right) \hat{\ell}_{z_i'}^a\right] \leq \sum_t N_t,$$

since $t$ is not part of the enumeration $z_j'$, so the two expectations are taken independently: $\mathbb{E}[\hat{\ell}_{z_j'}^a] \le 1$ and $\mathbb{E}[\hat{\ell}_t^a] \le 1$. Additionally we use that $q^a$ is a distribution. We now note that summing over $t$ or $s$ is equivalent in the above, i.e.,

$$\sum_t N_t \le \sum_t |\{s : s + d_s \in [t, t + d_t)\}| = \sum_s |\{t : s + d_s \in [t, t + d_t)\}|,$$

since counting in how many intervals every loss is observed in is the same as counting how many losses are observed in every interval. Note that we implicitly restrict both $s$ and $t$ to be in $[T]$.

We now split this

$$\sum_s |\{t : s + d_s \in [t, t + d_t)\}| = \sum_s |\{t > s : s + d_s \in [t, t + d_t)\}|$$
$$+ |\{t < s : s + d_s \in [t, t + d_t)\}|$$

and bound the first term as

$$|\{t > s : s + d_s \in [t, t + d_t)\}| \le |\{t > s : t \le s + d_s\} \setminus \{t > s : t + d_t < s + d_s\}|$$
$$\le d_s - |\{t > s : t + d_t < s + d_s\}|, \tag{22}$$

The second term is similarly bounded as

$$|\{t < s : s + d_s \in [t, t + d_t)\}| \le |\{t < s : s + d_s < t + d_t\}|. \tag{23}$$

Finally we note that by the prior equivalency of summing over $t$ or $s$, the negative term in (22) cancel with (23) once summed. This bounds the final term of (21) by $D$ and results in

$$\bar{\mathcal{R}}_T \le \frac{\ln K}{\eta'} + \eta'\left(\frac{eKT}{2} + D\right). \tag{24}$$

We now consider the truncation of the learning rate which is mandated by Lemma 11. If the input learning rate fulfils $\eta \le (2eN)^{-1}$ then $\eta' = \eta$, and (24) simply becomes

$$\bar{\mathcal{R}}_T \le \frac{\ln K}{\eta} + \eta\left(\frac{eKT}{2} + D\right),$$

where $\eta$ is the input learning rate.

If instead the learning rate is truncated, meaning the input learning rate is larger than $(2eN)^{-1}$, the algorithm uses $\eta' = (2eN)^{-1}$, meaning (24) becomes

$$\bar{\mathcal{R}}_T \le 2eN \ln K + \frac{\frac{eKT}{2} + D}{2eN} \le 2eN \ln K + \eta\left(\frac{eKT}{2} + D\right).$$

Taking the maximum of these two regret bounds finalizes the proof for any input learning rate $\eta$. $\quad\square$

## C Additional proofs

### C.1 Proof of Lemma 2

Consider first skipping just one round $s$. We then have

$$\bar{\mathcal{R}}_T := \mathop{\mathbb{E}}_{A_1,\ldots,A_T}\left[\sum_t \sum_a p_t^a \ell_t^a\right] - \min_a \sum_t \ell_t^a$$

$$\le \mathop{\mathbb{E}}_{A_1,\ldots,A_T}\left[\sum_a p_s^a \ell_s^a\right] - \min_a \ell_s^a + \mathop{\mathbb{E}}_{A_1,\ldots,A_T}\left[\sum_{t \ne s} \sum_a p_t^a \ell_t^a\right] - \min_a \sum_{t \ne s} \ell_t^a$$

$$\le 1 + \mathop{\mathbb{E}}_{A_1,\ldots,A_{s-1},A_s,\ldots,A_T}\left[\sum_{t \ne s} \sum_a p_t^a \ell_t^a\right] - \min_a \sum_{t \ne s} \ell_t^a$$

$$= 1 + \bar{\mathcal{R}}_{T \setminus \{s\}},$$

where the first inequality uses $\min_a[x_a + y_a] \ge \min_a x_a + \min_a y_a$ for any $x, y$ and the second inequality uses $\ell_s^a \in [0, 1]$ for all $a$ and $p_s^a$ being a distribution. In this line we also use the fact that no $p_t$ depend on $A_s$. The proof is then complete by iterating this argument over all $s \in C$.

## C.2  Proof of Lemma 5

The first inequality follows directly from insertion of $\beta_m = \sqrt{\omega_m}/4e \ln K$ into (4) and using the doubling condition for staying in the epoch (6).

For the second condition, we consider several cases of the optimal value in epoch $m$:

**Case 1**   If $\beta_m^* \geq \beta_m$ we have

$$\text{Bound}_m(\beta_m^*) \geq 4e\beta_m^* \ln K \geq 4e\beta_m \ln K = \sqrt{\omega_m} \geq \frac{\text{Bound}_m(\beta_m)}{3}, \tag{25}$$

In all following cases we consider $\beta_m^* < \beta_m$.

**Case 2**   We now consider the case where $\beta_m^* < \beta_m$ and the doubling happened because the number of skipped rounds grew to large. This implies the following inequality

$$\left(|S_{\beta_m}^m| + 1\right)^2 \geq \omega_m,$$

leading to

$$\text{Bound}_m(\beta_m^*) \geq |S_{\beta_m^*}^m| \geq |S_{\beta_m}^m| \geq \sqrt{\omega_m} - 1 \geq \frac{\text{Bound}_m(\beta_m)}{3} - 1, \tag{26}$$

where the second inequality comes from the assumption that $\beta_m^* < \beta_m$, meaning at least as many delays are skipped using $\beta_m^*$, as this is a lower threshold for skipping.

**Case 3**   If $\beta_m^* < \beta_m$ and the doubling instead happened because the second term grew too large, we have the following inequality:

$$\left(Ke/2 \cdot (\sigma(m) + 1) + D_{\beta_m}^m + \beta_m\right) \ln K \geq \omega_m. \tag{27}$$

In this case we have

$$\begin{aligned}
\text{Bound}_m(\beta_m^*) &\geq \frac{eK\sigma(m)/2 + D_{\beta_m^*}^m}{\beta_m^*} \\
&= \frac{\beta_m}{\beta_m^*}\left(\frac{eK\sigma(m)/2 + D_{\beta_m}^m}{\beta_m} + \frac{D_{\beta_m^*}^m - D_{\beta_m}^m}{\beta_m}\right) \\
&= \frac{\beta_m}{\beta_m^*}\left(\frac{eK\sigma(m)/2 + D_{\beta_m}^m}{\beta_m} - \Delta|S|\right) \tag{28} \\
&= \frac{\beta_m}{\beta_m^*}\left(4e\sqrt{\omega_m} - \frac{eK}{2\beta_m} - 1 - \Delta|S|\right) \tag{29} \\
&= \frac{\beta_m}{\beta_m^*}\left(4e\sqrt{\omega_m} - \frac{2e^2K\ln K}{\sqrt{\omega_m}} - 1 - \Delta|S|\right), \tag{30}
\end{aligned}$$

where (28) uses $D_{\beta_m}^m \leq D_{\beta_m^*}^m + \beta_m \Delta|S|$ for $\Delta|S| = |S_{\beta_m^*}^m| - |S_{\beta_m}^m|$. For (29) we use (27) with $\beta_m = \sqrt{\omega_m}/4e \ln K$.

Again we consider cases, this time of (30). Assume first

$$4e\sqrt{\omega_m} - \frac{2e^2K\ln K}{\sqrt{\omega_m}} - 1 - \Delta|S| \geq 2e\sqrt{\omega_m}.$$

which means

$$\text{Bound}_m(\beta_m^*) \geq \frac{\beta_m}{\beta_m^*}2e\sqrt{\omega_m} \geq \text{Bound}_m(\beta_m).$$

If we instead assume

$$4e\sqrt{\omega_m} - \frac{2e^2K\ln K}{\sqrt{\omega_m}} - 1 - \Delta|S| \leq 2e\sqrt{\omega_m},$$

which implies

$$\Delta|S| \geq 2e\sqrt{\omega_m} - \frac{2e^2 K \ln K}{\sqrt{\omega_m}} - 1,$$

we directly have

$$\text{Bound}_m(\beta_m^*) \geq \Delta|S| \geq \text{Bound}_m(\beta_m) - 2e^2 K \ln K - 1, \tag{31}$$

where we have used $\sqrt{\omega_m} \geq 1$. Note that this final inequality is the worst case of case 3.

Finally we compare the cases: Noting that they are exhaustive and by comparing (25), (26) and (31) the lemma is proven. $\qquad\square$

## C.3   Proof of Theorem 6

The idea of the proof is to use the nature of the doubling schema in the usual way, combined with Lemma 5 for the second to last epoch.

Let $M$ be the total number of epochs in the doubling schema:

$$\begin{aligned}
\bar{\mathcal{R}}_T &\leq \sum_{m=}^{M} \text{Bound}_m(\beta_m) \\
&\leq \sum_{m=1}^{M} 3\sqrt{\omega_m} \\
&= \sum_{m=1}^{M} 3\sqrt{2}^m \\
&= 3\frac{\sqrt{2}^{M+1} - 1}{\sqrt{2} - 1} \\
&\leq \frac{6}{\sqrt{2} - 1}\sqrt{2^{M-1}} \\
&= \frac{6}{\sqrt{2} - 1}\sqrt{\omega_{M-1}} \\
&\leq \frac{2}{\sqrt{2} - 1}\left(3\,\text{Bound}_{M-1}(\beta_{M-1}^*) + 2e^2 K \ln K + 1\right).
\end{aligned}$$

The proof is finalised by

$$\text{Bound}_{M-1}(\beta_{M-1}^*) \leq \min_{\beta}\left\{|S_\beta| + 4e\beta \ln K + \frac{KT + D_\beta}{4e\beta}\right\}. \quad \square$$