[Reviews · NeurIPS 2019]

Reviewer 1



The paper deals with algorithms and regret guarantees for the non-stochastic delayed reward bandit problem. The authors make three main contributions. 1. For the setting of non-stochastic bandit problems with unknown, variable, but bounded delays the authors establish regret guarantees for the delayed EXP3 algorithm. These regret guarantees establish a conjecture of Cesa-Bianchi[2016]. 2. The authors then study regret guarantees for the case where the delays are potentially unbounded, and variable. For this setting the authors provide an algorithm that is a slight variant of delayed EXP3. Regret bounds are established for this setting. 3. Furthermore, via a non-standard application of the doubling trick the algorithms are made robust to various unknown parameters. The results are significant and the problem of bandit learning with delayed feedback is significant. The writing could have been improved as it was not always easy to follow the paper. I have a few suggestions and questions. 1. It is important to clearly explain the setup early on in the paper, ideally in the introduction itself. I am not that familiar with delayed reward setup in bandits and it took me a bit of time to understand (1) How the regret of the learning algorithm is calculated. (2) What extra information does an oracle comparator have that the learning algorithm does not have. (3) How the delays effect the learning algorithm. Furthermore, I initially assumed that the delays are themselves arm dependent but that turned out not to be the case. Hence, it is important that the authors clearly explain the above points in the introduction so that the learning setup is clear. 2. It looks like the idea of stability-span is new to the analysis and that the definition/usage of this quantity is also a contribution of this paper. Can the authors confirm this? 3. On line 119, the authors claim that N \leq 2 \max d_t. However, since the set N_t has at most d_t points, it should be the case that N \leq \max d_t. Can the authors explain why a factor of 2 is necessary? 4. Does Theorem 1 assume that the losses are bounded? 5. For the skipper algorithm it would be great if the authors could provide some intuition as to why it achieves potentially lower regret. What would be especially insightful is if the authors can explain why the algorithm works fine even though it potentially throws away information. I understand that the stability span in this case is lower, but I would like to see a simple, explaination that does not have to resort to the use of stability span. 6. In section 4, it would be nice if the authors explained the reasoning behind the definition of epochs as shown in equations 4-6. For example, why is the definition of epoch in equation 6 the correct definition? Overall, I like this paper and I think it is an important contribution. However, the writing is not up to the mark and with some effort this paper can be made reader-friendly.

Reviewer 2



The paper is well-motivated especially the delay at action/observation time settings. These are real settings one must consider in practice and no previous works sufficiently studies them. My only real consider is the assumptions needed in this work. How reasonable are these in practice? Though, since this seems to be the first work in this area, I guess assumptions are OK to show initial results. The authors do acknowledge this and are able to come up with a refinement which removes assumptions. One compliant I have about the paper is that the authors do not provide an empirical results. The theoretical results are nice but this paper's quality would benefit from even the most rudimentary experimental section. I'm disappointed the authors did not include this section. ********** I have read the author's response and am not satisfied. The first point they make about not running experiments because it's not possible to cover all scenarios is a disappointing. Experiments do not have to cover all scenarios, that's kind of their empirical nature. There are plenty of scenarios the authors could experiment with to at least provide some additional evidence the algorithm works. The second point is even more disappointing. One always has something to compare against, otherwise how do you know your method has any merit? For example, compare against an algorithm which picks random arms. This is the minimum baseline and is always available to compare against. One can also compare against algorithms not designed for delays. If the proposed algorithm is worth publishing then it should beat these algorithms. I expect more from authors when submitting to a venue of this caliber.

Reviewer 3



The paper considers a version of the adversarial stochastic multi-armed bandit problem where the feedback received from pulling an arm is delayed. The delays are time-varying and can be selected by an oblivious adversary. The main algorithm presented is referred to as the “skipper” algorithm, and refines exponential weights analysis by skipping rounds with large delays, and is based on prior knowledge of the cumulative delay. When delays are only revealed each times an action is taken, and no prior information is granted on the cumulative delay (or the horizon length) the authors provide a doubling scheme that achieves a refined regret bound. I like the paper and definitely appreciate various aspects of it. The topic and the algorithmic approach are interesting, the paper is well-written, and, as far as I could tell, analytically correct. I don't have major concerns regarding the paper. My main comments are as follow. 1. Since in many application domains delay information is only realized when feedback arrives. This is true also in the motivating examples that are given by the authors in page 2 (while expected delay might be available at time of action, the realized delay can often be different from this expectation. It would definitely be insightful to understand how one could extend current results and algorithms in that direction, and a first step in this direction can be to have some robustness analysis that will help to understand to extent to which the performance of the doubling scheme deteriorates when actual delays are (even a bit) different from the one observed at the time of the action. 2. Similarly, it would be insightful to understand how robust the skipper algorithm is for misspecifications in the cumulative delay D. I apologize if I missed a discussion along these lines in the paper. 3. It would be very insightful, and would clearly increase the contribution of the paper, to have some form of optimality for the results. Ideally that would be established through a refined lower bound for time-varying delays, but even some discussion using examples or numeric analysis might be helpful in that respect.

[Author Response · NeurIPS 2019]

# Author Response for Submission 3526

We thank the reviewers for their effort, thoroughness, and constructive comments.

**To Reviewer 1**

1. We appreciate your suggestions for improving the presentation and will follow them in the final version.

2. Stability-spans have been previously used in the full information setting by Joulani et al. (2016). (Joulani et al. denote the quantity by $\tilde{\tau}$, but do not give it an explicit name.) The use of stability-spans in the analysis of delayed Exp3 is new and generalizes the role of the delay in the fixed-delay setting (Cesa-Bianchi et al., 2016).

3. The stability-span $N_t$ is the amount of feedback that arrives between playing action $A_t$ and observing its feedback. This may include up to $\max_s d_s$ observations from the actions that were played *before* $A_t$ (assuming that their delay is large enough, so that they arrive after time $t$) *and* up to $d_t$ observations from actions that were played *after* $A_t$ (assuming that their delay is small enough, so that they arrive before $t + d_t$). Together it gives the factor of 2.

4. Regarding bounded losses in Theorem 1: We assume that the losses are in the unit interval $[0, 1]$, which is a customary assumption in many bandit papers. We will make sure to state this explicitly.

5. By throwing away information from observations with excessively large delays we obtain a *simpler analysis* of the algorithm. We do not claim that throwing away information lowers the regret. The analysis of weight updates for observations with large delays requires stability of the weights over the corresponding time span. When the delays are highly unequal, from the analysis perspective it is cheaper to ignore the large delays than to analyze them. As long as the number of skipped observations is comparable with the regret bound for the remaining rounds, we do not lose much in the regret bound (at most a constant factor), but significantly simplify the analysis.

6. The reasoning behind the definition of the epochs is to balance the individual terms of the bound in equation (4). The selection of $\beta_m$ in equation (5) directly controls the middle term, while the doubling condition in equation (6) makes sure that the sum of the first and the last terms is of the same order as the middle term. We will add the intuition to the final version of the paper.

**To Reviewer 2**

1. Regarding experiments: We agree that in general experiments are a valuable addition to corroborate theoretical results, however, there are a number of reasons that make it difficult to design comprehensive experiments for our work. First of all, it is impossible to design comprehensive experiments for algorithms for adversarial problems because of the impossibility to cover all possible adversarial scenarios. Second, this is the first work on adversarial bandits with arbitrary delays and we had no natural prior work to compare to. We believe that adding experiments at this stage would constitute an overly major change, but if the reviewer has any particular setups in mind (what kind of loss sequences and delays should we test; what algorithms should we compare to) we will be happy to consider them in potential extensions of the work.

**To Reviewer 3**

1&2. The proposed ideas for extension of our work are very interesting! In particular, the robustness analysis and the idea of receiving the expected regret at action time and the realized regret at observation time would be an interesting variation of the problem. This would relax the assumption of "observation at action" time significantly. We believe that it should be possible to achieve regret guarantees without prior knowledge of $T$ and $D$ in this setting, something that has not yet succeeded in the harder "delay at observation time" setting.

3. As mentioned in our discussion, refined lower bounds for varied delays would be incredibly interesting. As we have written in the paper, our results match the lower bound up to logarithmic factors in the case of uniform delays. It is also easy to see that we match the lower bound up to logarithmic factors in the other extreme case described in Example 8: when observations for $\mathcal{O}(\sqrt{KT})$ rounds arrive at the end of the game and observations for the remaining rounds arrive with no delay. In this case there are $\Omega(T)$ no-delay rounds and we have the standard $\Omega(\sqrt{KT})$ lower bound for the no-delay rounds by the standard multiarmed bandits analysis (Auer et al., 2002), which implies the same lower bound for the whole game. This lower bound is matched by our algorithm within logarithmic factors, as described in Example 8. It does not seem trivial to obtain lower bounds for intermediate setups between the two extremes and we leave it to future work. We will add the discussion above to the paper.

[Meta-Review · NeurIPS 2019]

This paper significantly extends previous work in multi-armed bandits with delayed feedback by allowing for unbounded delays. While theoretical, this work should be practically relevant as it only uses bandit feedback, it avoids making stochastic assumptions on the losses, and (of course) it allows for delayed feedback, the latter being needed to capture many real-world problems. This paper generated a good amount of discussion, so I’ll mention the pros and cons. On the positive side, as mentioned above, this paper significantly extends previous work in multi-armed bandits with delayed feedback by allowing for unbounded delays. As the first paper to consider bandits with unbounded delays, an important restriction in all previous works has been relaxed, and given that this work is for non-stochastic bandits, there could be many practical applications. Additionally, the algorithmic approach is interesting and original, and the proofs appear to be technically correct. On the (somewhat) negative side, the way in which unbounded delays are handled is less than ideal. For one result, the total (cumulative) delay D needs to be known. For another result which avoids needing to know D, the delay is known at the time of pulling the arm. Relaxing this assumption to expected delay (or not needing to know this delay until observation time while still adapting to D and T) would be better. That said, the paper’s results and analysis still appear to be technically sophisticated, so it would be fine to leave these potential improvements for the future. One controversial negative aspect was the lack of experiments, about which one reviewer was very disappointed. However, this reviewer also expressed low confidence for their review, and I felt that the paper can stand on its theory; moreover, doing meaningful experiments (with meaningful baselines) is quite challenging here. While i.i.d. equal mean sequences can work well in the vanilla non-stochastic setting, in the delayed feedback setting this becomes less clear. This work will be a welcome addition to the proceedings. I ask the authors to please improve the writing / presentation as per Reviewer 1’s suggestions, as Reviewer 1 argued for acceptance assuming these would be improved.